# Application of non-invasive low-intensity pulsed electric field with thermal cycling-hyperthermia for synergistically enhanced anticancer effect of chlorogenic acid on PANC-1 cells

Chueh-Hsuan Lu[1,2], Yu-Yi Kuo[1,2], Guan-Bo Lin[1,2], Wei-Ting Chen[1,2], Chih-Yu Chao[1,2,3]*

1 Department of Physics, Lab for Medical Physics & Biomedical Engineering, National Taiwan University, Taipei, Taiwan, 2 Biomedical & Molecular Imaging Center, National Taiwan University College of Medicine, Taipei, Taiwan, 3 Institute of Applied Physics, National Taiwan University, Taipei, Taiwan

* cychao@phys.ntu.edu.tw

**Data Availability Statement:** All relevant data are within the manuscript and its Supporting Information files.

## Abstract

Most existing cancer treatments involve high-cost chemotherapy and radiotherapy, with major side effects, prompting effort to develop alternative treatment modalities. It was reported that the combination of thermal-cycling hyperthermia (TC-HT) and phenolic compound exhibited a moderate cytotoxic effect against human pancreatic cancer PANC-1 cells. In this study, we investigate the efficacy of triple combination in PANC-1 cancer cells by adopting low-intensity pulsed electric field (LIPEF) to couple with TC-HT and CGA (chlorogenic acid). The study finds that this triple combination can significantly impede the proliferation of PANC-1 cells, with only about 20% viable cells left after 24h, whereas being non-toxic to normal cells. The synergistic activity against the PANC-1 cells was achieved by inducing G2/M phase arrest and apoptosis, which were associated with up-regulation of p53 and coupled with increased expression of downstream proteins p21 and Bax. Further mechanism investigations revealed that the cytotoxic activity could be related to mitochondrial apoptosis, characterized by the reduced level of Bcl-2, mitochondrial dysfunction, and sequential activation of caspase-9 and PARP. Also, we found that the triple treatment led to the increase of intracellular reactive oxygen species (ROS) production. Notably, the triple treatment-induced cytotoxic effects and the elevated expression of p53 and p21 proteins as well as the increased Bax/Bcl-2 ratio, all could be alleviated by the ROS scavenger, N-acetyl-cysteine (NAC). These findings indicate that the combination of CGA, TC-HT, and LIPEF may be a promising modality for cancer treatment, as it can induce p53-dependent cell cycle arrest and apoptosis through accumulation of ROS in PANC-1 cells.

**Funding:** This work was supported by grants from Ministry of Science and Technology (MOST 108-2112-M-002-016 to CYC) and Ministry of Education (MOE 106R880708 to CYC) of the Republic of China. The funders had no role in study design, data collection and analysis, decision to publish, or preparation of the manuscript.

**Competing interests:** The authors have declared that no competing interests exist.

## Introduction

Cancer has been a leading cause of mortality worldwide [1]. Major options for cancer treatment nowadays involve chemotherapy, radiation, and surgery, which, however, entail serious side effects and high costs, on top of high risk of tumor recurrence and refractoriness [2]. Therefore, there is a need for new approaches in cancer treatment which are more efficient and economically feasible. Like many other human diseases, cancer is driven by a complex network of gene-environment interactions [3]. Therefore, a prerequisite for new cancer treatments is the capability to target multiple pathways.

In the past several decades, combination therapies, especially combination drugs, have been a major subject in oncology due to their synergistic or additive anticancer effects [4]. Although combination therapies show a benefit over monotherapy, they could be toxic if chemotherapeutic drug is used as one of agents. Moreover, there still exist potentially harmful effects resulting from drug-drug interactions [5]. Therefore, feasibility for the use of physical stimuli as a therapeutic agent for cancer treatment has been explored [6]. The physical stimuli not only have potential advantages of providing non-invasiveness and safety, but also can be controlled in a precise manner to obtain a non-harmful means toward non-cancerous cells. We have recently shown that the thermal cycling-hyperthermia (TC-HT) combined with chlorogenic acid (CGA) has an anticancer synergy in combating human pancreatic cancer PANC-1 cells [7], but the effect is moderate. On the other hand, previously, the incorporation of pulsed electric field (PEF) into triple remedy has been demonstrated to potentiate the cytotoxicity of double combination of ultrasound and herb by our group [8]. Therefore, combination of physical stimuli for improving the anticancer ability of natural compound appears to be a promising new treatment for cancer.

Anticancer therapies with electricity-based methods have been found to directly kill cancer cells or to enhance drug delivery and absorption [9, 10]. Among them, the administration of electric current is the most typical modality. Irreversible electroporation is a treatment that employs high-voltage (mostly several kV/cm) pulsed electric current for tumor ablation, mainly by inducing necrotic process through direct contact of electrodes [11]. Another tumor ablation modality with electric stimulation for drug administration is electrochemotherapy, which combines the non-permeant or low-permeant chemotherapeutic drugs and the reversible membrane electroporation induced by high-voltage pulses (over hundreds of V/cm to one kV/cm) [12]. However, the employment of electricity with such a strong intensity may cause serious problems, such as cardiac arrhythmias and muscle contraction [13]. Moderate intensity (0.45 and 0.63 kV/cm) electroporation in presence of calcium has also been demonstrated to be capable of inducing cancer cell death [14]. It should be noted that even for some proposed approaches using low-voltage (<200V/cm) electroporation to increase the permeability of cell membranes, electrolysis has been reported to the cause of cytotoxic effects resulting from the direct contact with electrodes [15]. Therefore, we suggest that the electrical stimulation in low-intensity and non-invasion manner would be more proper for anticancer treatment. Non-invasive PEF has the merit of generating neither conduction current nor Joule effect in the administration of electric field. As a viable method in the application of physical stimuli, non-invasive PEF features external application of electric pulse to the desired area. In practice, the use of non-invasive PEF to enhance the activity of anticancer agent and lower the required dosage of agent is significantly effective, as proven by our recent studies on polyphenols with this approach [16, 17]. It was found that exposure to non-invasive, low-intensity PEF (LIPEF) has strengthened the effect of low-dose curcumin in combating human pancreatic cancer PANC-1 cells [16]. Our team also observed the consistent effect of non-invasive LIPEF in significantly enhancing EGCG activity in human pancreatic cancer cells [17]. It is

noteworthy that LIPEF alone did not harm the cells, underscoring its safety and non-toxicity. Therefore, combination of LIPEF with herbal agent or physical stimulus appears to be a viable approach for cancer treatment. The study, therefore, focuses on the effect of combining LIPEF with CGA and TC-HT in enhancing the therapeutic efficacy in cancer treatment.

In this paper, we study the effect of triple combination of non-invasive LIPEF and TC-HT stimuli with herbal agent CGA on growth inhibition of PANC-1 cells and p53 signalling pathway. The result shows that the triple treatment can produce a significant synergistic cytotoxicity in human pancreatic cancer PANC-1 cells. The benefits of this triple combination can be attained via cell cycle arrest and intrinsic apoptosis resulting from ROS accumulation. The processes are found to be associated with the increase in the levels of p53, p21, and Bax as well as the decrease in the level of Bcl-2, accompanied subsequently with the activation of caspase-9 and cleavage of PARP triggering apoptosis. The findings point out that combining the non-invasive LIPEF and TC-HT along with CGA may be a potent strategy for anticancer treatment.

## Materials and methods

### Cell culture

Human pancreatic cancer cell line PANC-1 and AsPC-1 were obtained from the Bioresource Collection and Research Center of the Food Industry Research and Development Institute (Hsinchu, Taiwan). Normal human pancreatic duct cell line H6c7 was obtained from Kerafast, Inc. PANC-1 and AsPC-1 cells were cultured respectively in high-glucose DMEM and RPMI-1640m medium (Hyclone) supplemented with 10% fetal bovine serum (Hyclone) and 1% penicillin-streptomycin (Gibco). H6c7 cells were cultured in keratinocyte-serum free medium (Invitrogen) supplemented with human recombinant epidermal growth factor, bovine pituitary extract (Invitrogen), and 1% penicillin-streptomycin. Both cell lines were maintained in a humidified 5% $CO_2$ incubator at 37°C.

### Experimental setup of LIPEF

There are two models of electrical stimulation: conductive coupling and capacitive coupling [18]. In our experimental design, we adopted capacitive coupling model to investigate the externally applied LIPEF in combination with anticancer agents on pancreatic cancer. As shown in **Fig 1A** and **1B**, two copper parallel electrodes separated by an air gap present a non-contact means to build up an electric field passing through biological sample, and the cells seeded in a culture well were placed between two flat and parallel electrodes. The distinct electrical stimulation of LIPEF is characterized by a pulse train waveform with low electric field intensities ($< 100$ V/cm), a pulse frequency of 2 Hz, and a pulse duration of 2 ms (**Fig 1C**). The method of such non-invasive strategy of LIPEF is somewhat similar to that of tumor treating fields [19], but in a non-contact approach with different waveform and field strength. Furthermore, exposure to non-invasive LIPEF can be viewed as the simultaneous interaction of multiple frequencies [20] with a biological sample due to the electrical dispersion caused by the encounter with biological dielectrics.

### Experimental setup of TC-HT

The experimental setup and administration of TC-HT (10-cycles) have been previously described [7] with optimum results. The TC-HT was performed by using a modified PCR system to achieve a series of short period of heat exposure within the desired time. In our experimental design of TC-HT, the protruded portion of culture well was cut off for better

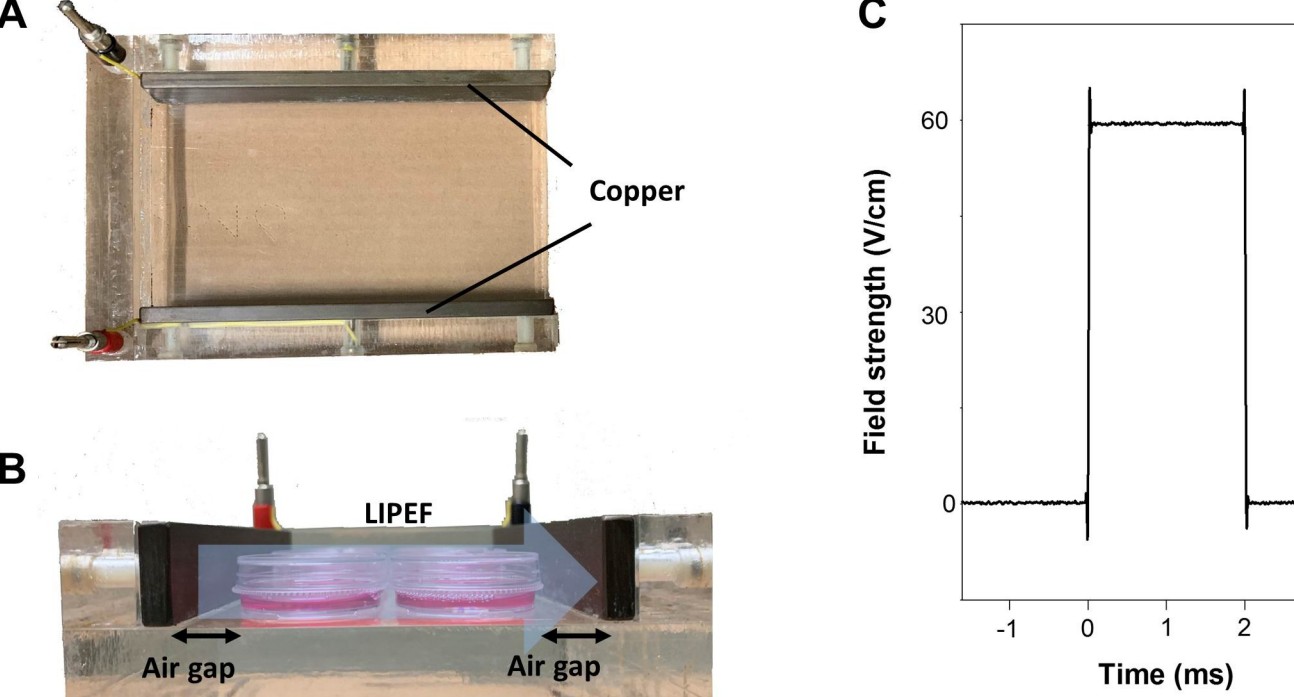

**Fig 1. Experimental device and waveform for the exposure to LIPEF.** (A) A photo of the electric field stimulation device. (B) The cells seeded in a culture well were located in the stimulation device constructed of two parallel copper plates. Electric fields of 30, 60, and 90 V/cm were introduced between electrodes. The light blue arrow shown here indicates the schematic representation of non-invasive LIPEF exposure on cells. (C) The waveform of 60V/cm LIPEF applied on the electrodes.

immersion in water bath consisting of the heat sink of PCR machine, and the temperature experienced by cells was measured by a needle thermocouple. The PANC-1 cells were subjected to the cycling temperature from 43.5 to 36°C. In order to return the temperature of the cancer cells seeded in culture well to physiological temperature after each thermal exposure, active cooling environment is needed to be involved for enabling such a bulk (culture well) to dissipate heat quickly *in vitro*.

## CGA, TC-HT, LIPEF and combined treatment

The CGA were dissolved in dimethyl sulfoxide (DMSO) (Sigma) and stored at -20°C. The stocks were diluted with a culture medium to the indicated final concentration for treatment before usage. Cells were incubated overnight at 37°C and then treated with various concentrations of the CGA. For the double treatments of CGA and physical stimulus, CGA-treated cells were subjected to TC-HT or LIPEF. The double treatment of physical stimuli was performed by exposing the cells to the TC-HT first and then to the LIPEF. In the triple treatment, cells in a medium containing CGA were exposed to the TC-HT followed by the LIPEF. The timeline of combined treatment procedures is shown in **Fig 2A**.

## Cell viability

Cell viability was accessed by 3-(4,5-dimethylthiazol-2-yl)-2,5-diphenyltetrazolium bromide (MTT) (Sigma) assay. Cells were seeded in 24-well plates and incubated overnight at 37°C. After treatment with CGA, TC-HT, and LIPEF alone or in combination for 24 h, the medium

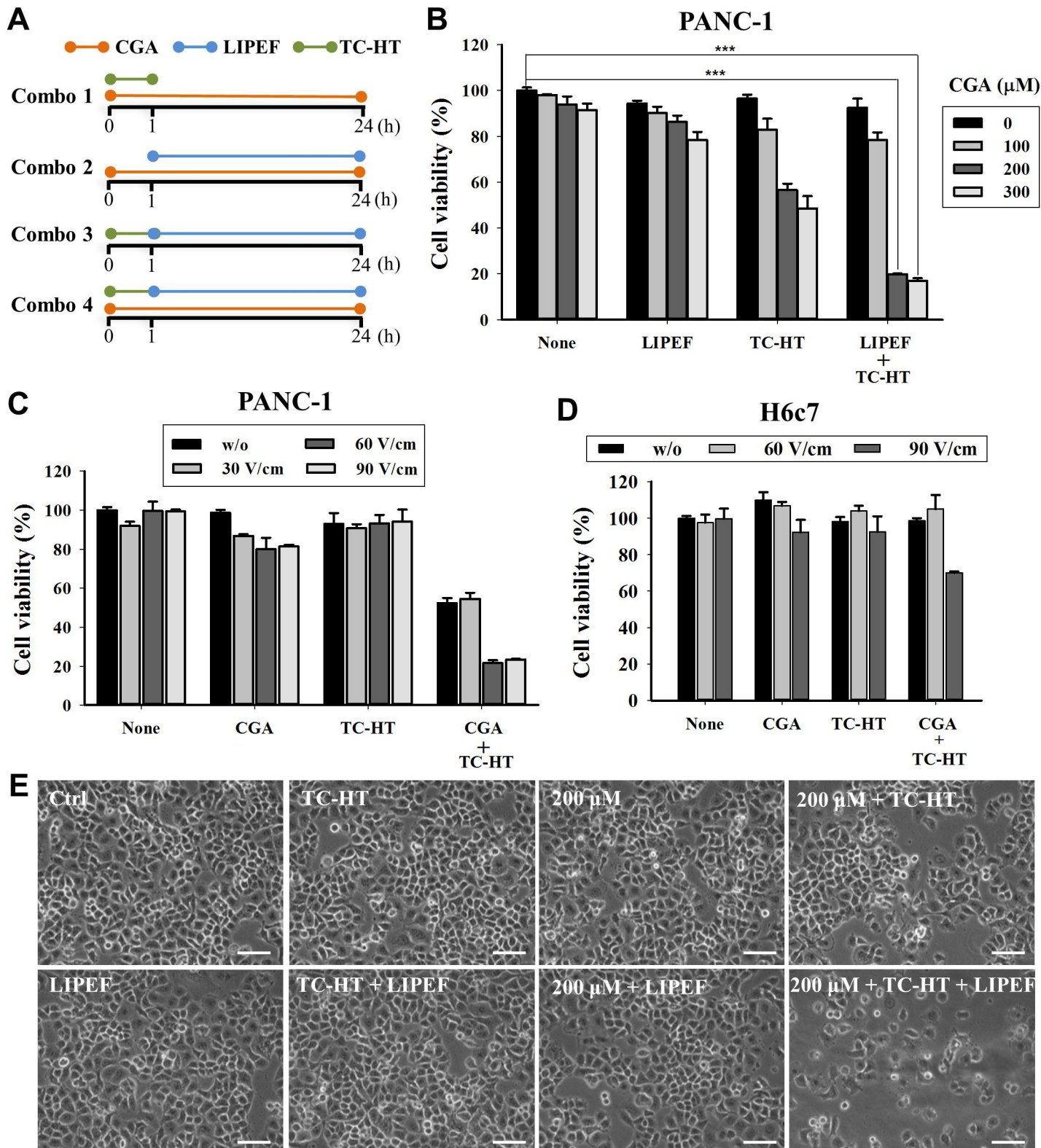

**Fig 2. Viability and morphological changes of PANC-1 cells.** (A) Schematic of experimental design and timeline for combined treatments. Combo 1: CGA + TC-HT; Combo 2: CGA + LIPEF; Combo 3: TC-HT + LIPEF; Combo 4: CGA + TC-HT + LIPEF. (B) Cells were treated with CGA (100, 200, and 300 μM), TC-HT (10-cycles), and LIPEF (60 V/cm) alone, in double or triple combination for 24 h, and then cell viability was determined by MTT assay. (C) The viability of PANC-1 cells subjected to 200 μM CGA, 10-cycles TC-HT, and different intensities of LIPEF. (D) The viability of H6c7 cells was evaluated by MTT assay after triple treatment with 200 μM CGA,

10-cycles TC-HT, and different intensities of LIPEF. (E) Representative images of morphological changes of PANC-1 cells under the light microscope after treatment with CGA (200 μM), TC-HT (10-cycles), and LIPEF (60 V/cm) alone or in combination for 24 h. Scale bar = 100 μm Data are presented as mean ± S.D. in triplicate. (***$p < 0.001$ vs. untreated control).

was removed and the cells were washed with phosphate buffered saline (PBS). Cells were then incubated in culture medium containing 0.5 mg/ml MTT for an additional 4 h at 37°C. DMSO was added to dissolve the formazan crystals and the absorbance was measured at 570 nm using an ELISA microplate reader. The calculation of synergism quotient (SQ) was dividing the combined effect by the sum of individual effects.

## Clonogenic survival assay

PANC-1 cells were seeded at 1000 cells/dish in 35 mm Petri dishes for 24 h and treated with CGA, TC-HT, and LIPEF alone or in combination. Cell medium was replaced after the treatment, and the dishes were cultured in a humidified 5% CO2 incubator at 37°C for additional 14 days. At last, the cells were fixed with 4% paraformaldehyde (Sigma) for 10 min and stained with 0.1% crystal violet (Sigma). The colonies containing more than 50 cells were counted, and the number of colonies in each treatment group was normalized to control group.

## Flow cytometric analysis of apoptosis

After single or combined treatment for 24 h, the apoptosis of PANC-1 cells was determined by using the Annexin V-FITC/PI apoptosis detection kit (BD Biosciences). The cells were harvested with trypsin-EDTA (Gibco) and collected by centrifugation at $2,000 \times g$ for 5 min, washed twice with cold PBS, and resuspended in binding buffer containing Annexin V-FITC and PI. The cell suspensions were incubated for 15 min at room temperature in the dark and analyzed by a FACS Calibur flow cytometer.

## Mitochondria membrane potential (MMP) measurement

The cells treated with CGA, TC-HT, and LIPEF for 24 h alone or in combination were collected, resuspended in PBS and incubated with 20 nM $DiOC_6(3)$ (Enzo Life Sciences International Inc.) for 30 min at 37°C in the dark. After $DiOC_6(3)$ staining, the fraction of cells showing low MMP was then measured by a flow cytometer.

## Cell cycle analysis

After 24 h treatment, the cells were collected by trypsinization and fixed in 70% ice-cold ethanol at 4°C overnight. Then, the cells were washed with cold PBS and treated with RNase A (0.1 mg/ml) for 20 min at 37°C. Finally, the cells were stained with PI (0.2mg/ml) for 30 min at room temperature in the dark. The DNA content of cells was then analyzed by flow cytometry.

## Measurement of ROS production

Cellular reactive oxygen species (ROS) levels of superoxide anion ($O_2^{\bullet-}$) were detected using the fluorescent dye dihydroethidium (DHE) (Sigma). In order to detect the ROS production induced by treatments, PANC-1 cells were treated with CGA, TC-HT, and LIPEF alone or in combination and then washed with PBS. The cells were incubated with 5 μM DHE for 30 min at 37°C in the dark. The fluorescence intensity was measured by flow cytometry, and ROS levels were expressed as mean fluorescence intensity (MFI) for comparison.

## Western blot analysis

PANC-1 cells were treated with CGA, TC-HT, and LIPEF for 24 h alone or in combination. The cells were harvested from each treatment, washed with cold PBS, and lysed on ice for 30 min in lysis buffer (Millipore). Cell lysates were then clarified by centrifugation at 23,000 × g for 30 min at 4°C, and the protein concentration in the supernatant fraction was quantified using the Bradford protein assay (Bioshop). Proteins were resolved by 10% SDS-PAGE and electrotransferred onto polyvinylidene fluoride membrane (Millipore) in transfer buffer (10 mM CAPS, pH 11.0, 10% methanol). The membranes were blocked with 5% nonfat dry milk/ TBST (blocking buffer) for 1 h at room temperature and then incubated overnight at 4°C with diluted primary antibodies in blocking buffer. The specific primary antibodies against Bcl-2, cleaved caspase-9 and Bax (Cell Signalling), p53, p21, cleaved PARP and GAPDH (GeneTex) were used. After washing with TBST, the membranes were incubated with HRP-conjugated anti-goat (GeneTex) or anti-rabbit (Jackson Immunoresearch) secondary antibody. Chemiluminescence was detected using WesternBright ECL western blotting reagent (Advansta). The intensities of bands were quantified by Amersham Imager 600 (AI600, GE Healthcare Life Science) and normalized to GAPDH, which served as loading control.

## Statistical analysis

The results were presented as mean ± standard deviation (SD) and performed in triplicate. Statistical analyses using one-way analysis of variance (ANOVA) were performed with SigmaPlot software. Differences with $p$-values less than 0.05 were considered to be statistically significant.

## Results

### Combination of LIPEF, TC-HT, and CGA effectively inhibits PANC-1 cell proliferation

To examine whether the LIPEF and TC-HT could enhance the anticancer activity of CGA, the inhibitory effect of single, double or triple combination of LIPEF, TC-HT, and CGA on PANC-1 cells growth was evaluated by MTT assay. In this study, as mentioned previously, we select the parameter 10-cycles for TC-HT and different field strengths of LIPEF in following experiments. The schematic of experimental procedure for double or triple treatment (CGA, TC-HT, and LIPEF) was shown in **Fig 2A**. In case of the groups including CGA, CGA was added at the beginning of treatments and the cells were exposed continuously to CGA. The results of MTT assay shown in **Fig 2B** demonstrated that there was no significant alteration in the cell viability between untreated control and the cells treated with TC-HT and 60 V/cm LIPEF alone or in combination. However, a notable dose-dependent reduction in cell viability was observed in double treatment containing CGA. Interestingly, the treatment with a triple combination of CGA (200 and 300 μM), TC-HT, and LIPEF (60 V/cm) displayed remarkable inhibition of PANC-1 cell proliferation, which shows nearly 80% decrease of cell viability as compared with the untreated group (**Fig 2B**). As can be seen from **Table 1**, the strongest

**Table 1. Synergy quotient for CGA in combination with TC-HT and LIPEF.**

| CGA (μM) | | TC-HT | LIPEF | TC-HT + LIPEF |
|---|---|---|---|---|
| | 0 | - | - | 0.82 |
| | 100 | 3.12 | 1.30 | 1.94 |
| | 200 | 4.48 | 1.16 | 5.22 |
| | 300 | 4.23 | 1.51 | 4.66 |

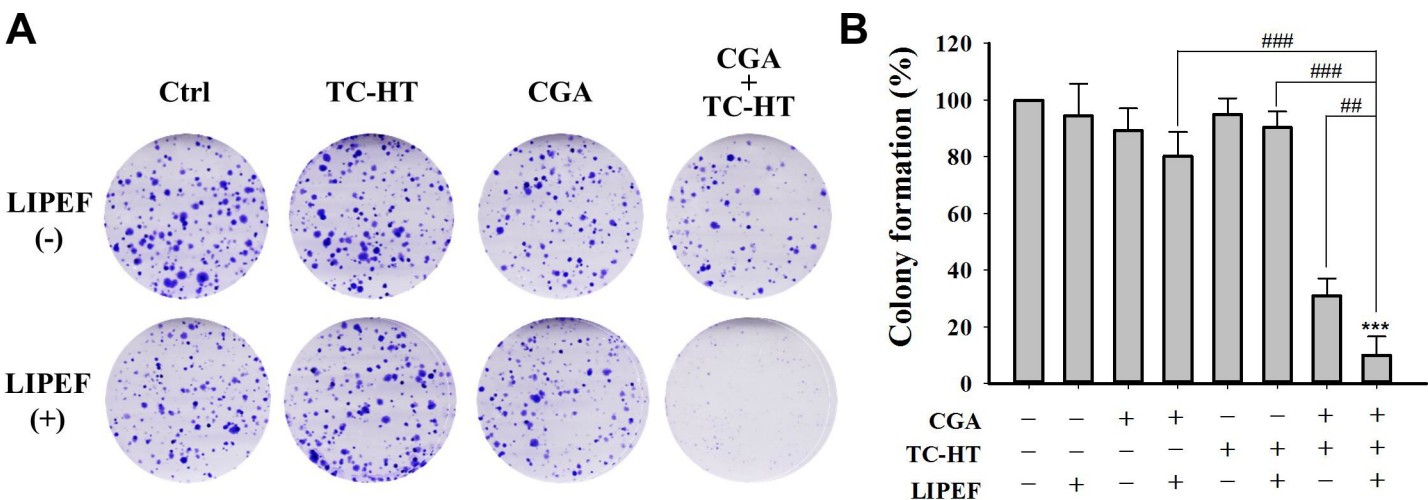

**Fig 3. Triple treatment suppresses colony formation of PANC-1 cells.** (A) Clonogenic survival assay results for PANC-1 cells treated with CGA (200 µM), TC-HT (10-cycles), and LIPEF (60 V/cm) alone, in double or triple combination. Images are representative of three independent experiments. (B) Analysis of colony formation rate. Data are presented as mean ± S.D. in triplicate. (***$p < 0.001$ vs. untreated control; ##$p < 0.01$ and ###$p < 0.001$ between indicated groups).

synergy of combining CGA with physical stimuli was obtained for the concentration of 200 µM CGA combined with TC-HT (10-cycles) and LIPEF (60 V/cm). Thus, a dose of 200 µM CGA was selected for subsequent analysis. We then performed the investigation of different LIPEF intensity (30, 60, and 90 V/cm) to study its effect on the triple treatment of PANC-1 cells and H6c7 pancreatic normal cells. Our result showed that the efficacy of double combination of TC-HT and 200 µM CGA in PANC-1 cells was enhanced at LIPEF intensity above 60 V/cm (**Fig 2C**). By contrast, the viability of H6c7 cells was not affected under the same conditions with 60 V/cm LIPEF but was reduced with 90 V/cm LIPEF (**Fig 2D**), suggesting that the triple treatment may cause specific cytotoxicity under certain threshold to cancer cells. Based on these observations, we chose 60 V/cm as an optimal parameter for LIPEF in subsequent experiments. The microscopic images shown in **Fig 2E** indicated significant morphological changes of the PANC-1 cells treated with CGA (200 µM) together with TC-HT (10-cycles) and LIPEF (60 V/cm) for 24 h such as shrunken and rounded shape and a reduced number of cells. Next, we continued to adopt 200 µM CGA and performed a clonogenic assay to confirm the effects of TC-HT (10-cycles) and/or LIPEF (60 V/cm) on CGA-treated cells. Our results revealed a marked suppressive effect on colony formation of PANC-1 cells subjected to triple combination treatment (**Fig 3A and 3B**), which is in agreement with the MTT cytotoxic data.

## Triple treatment induces G2/M cell cycle arrest in PANC-1 cells

To investigate the further features of proliferation inhibition by treatment with CGA (200 µM), TC-HT (10-cycles), and LIPEF (60 V/cm) in PANC-1 cells, we performed flow cytometric analysis of DNA content with PI. As shown in **Fig 4**, in the untreated PANC-1 cells, there were 60, 17, and 20% of cells in G1, S, and G2/M phases of the cell cycle, respectively. A similar cell cycle distribution profile was also obtained in the cells treated with either of CGA, TC-HT, and LIPEF. When LIPEF was co-treated with CGA or TC-HT, both of the double treatments still had no obvious effect on cell cycle progression. However, the combination of CGA and TC-HT led to an accumulation of cells in the G2/M phase (36.52%) with concurrent decline in the G0/G1 phase (45.50%) compared with control. Furthermore, a greater

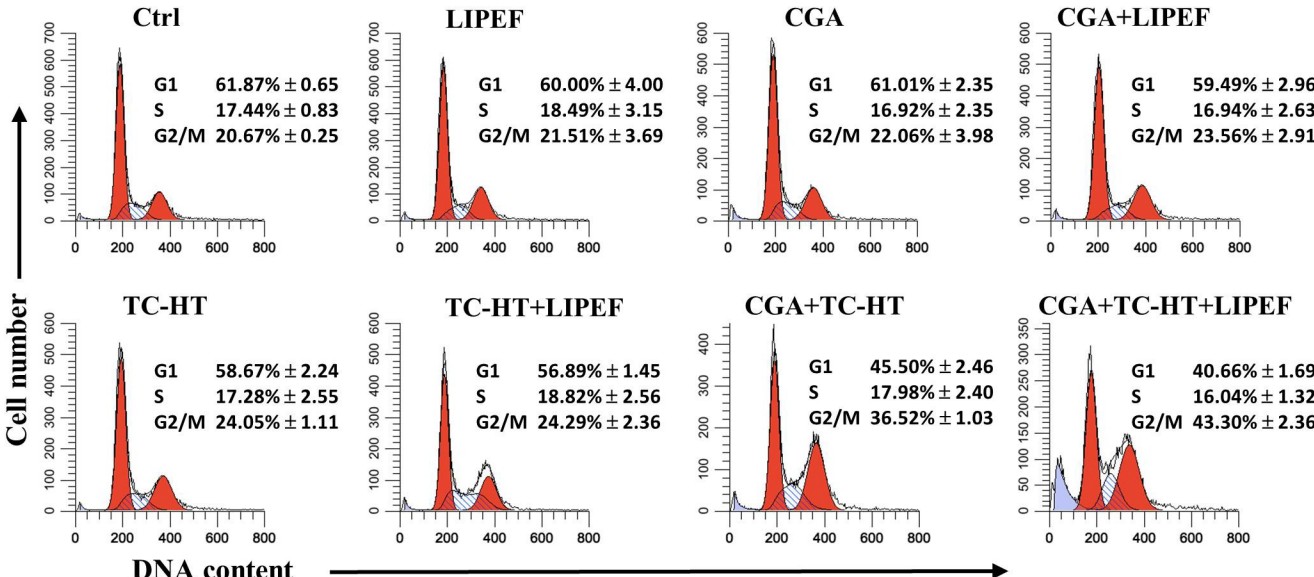

**Fig 4. Effects of CGA, TC-HT, and LIPEF on PANC-1 cell cycle distribution.** Cells were treated with CGA (200 μM), TC-HT (10-cycles), and LIPEF (60 V/cm) alone, in double or triple combination for 24 h, stained with PI and analyzed by flow cytometry for cell cycle analysis. Each histogram indicating the DNA content (x-axis) vs. cell number (y-axis) is representative of three independent experiments. The percentages of cells presented as mean ± S.D in G1, S, and G2/M phases are listed.

accumulation was observed in the PANC-1 cells following triple treatment exhibiting an apparent G2/M arrest (43.30%), and we also noticed an increase in the population of sub-G1 cells undergoing apoptosis in the CGA, TC-HT, and LIPEF triple treated group.

## LIPEF enhances the effect of the double treatment with CGA and TC-HT-induced apoptosis of PANC-1 cells

To investigate whether the synergistic suppressive effect of triple treatment was related to apoptosis, the quantitative analysis of PANC-1 cell apoptosis was measured by flow cytometry for all individual agents and their combinations. As shown in **Fig 5A and 5C**, there was minimal or no apoptosis induced in the cells exposed to single treatment or double treatment with TC-HT (10-cycles) and/or LIPEF (60 V/cm). Interestingly, it was found that cells doubly treated with CGA (200 μM) and physical stimulus displayed an increase of apoptotic rate, with 18% apoptosis for the LIPEF (60 V/cm) and, to a greater extent, 42.4% for the TC-HT (10-cycles) compared with control (4.7%). Noteworthy, when all three agents were administered together, the triple treatment induced a significantly higher proportion of apoptotic cells (66.7%) compared with each of the double combination or the control group.

## Effect of triple treatment on mitochondrial membrane potential (MMP) in PANC-1 cells

As depolarization of MMP was associated with apoptosis, we further examined the influence of each treatment on the change of MMP by flow cytometry using $DiOC_6(3)$ staining [21]. **Fig 5B and 5D** show that single treatment with either of CGA, TC-HT, and LIPEF alone or dual-physical stimuli with TC-HT and LIPEF did not alter the MMP. In response to the double treatment with CGA (200 μM) and LIPEF (60 V/cm), a moderate mitochondrial depolarization in PANC-1 cells was observed. Besides, the cells doubly treated with CGA (200 μM) and TC-HT (10-cycles) showed a notable reduction in MMP in comparison to the control cells.

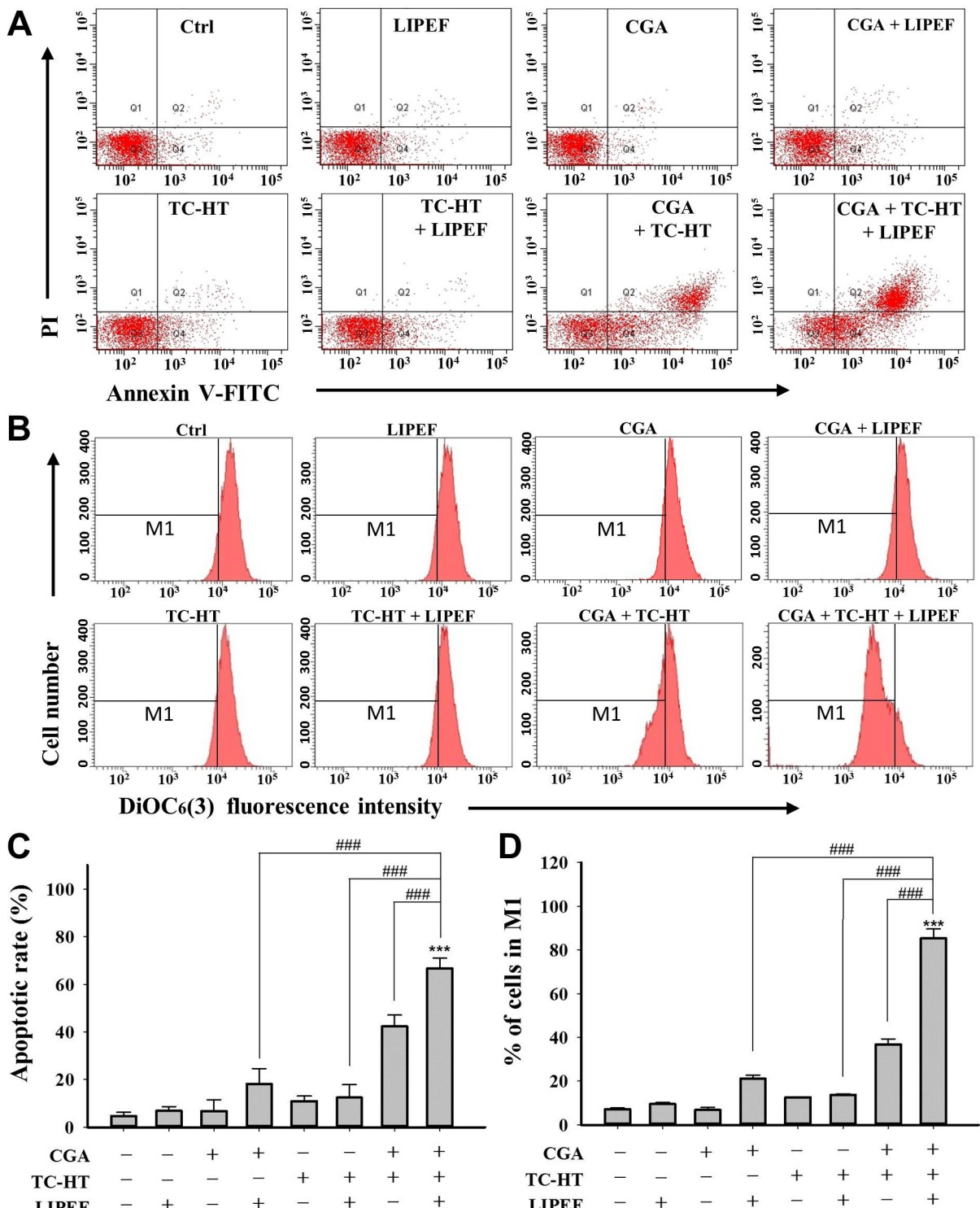

**Fig 5. Triple combination of CGA, TC-HT, and LIPEF induces PANC-1 cells apoptosis through mitochondrial pathway.** The apoptosis analysis for the PANC-1 cells following the treatment with CGA (200 μM), TC-HT (10-cycles), and LIPEF (60 V/cm) alone, in double or triple combination for 24 h. (A) Flow cytometric detection of the apoptosis with Annexin V-FITC/PI double staining. (B) Flow cytometric analysis of MMP using DiOC$_6$(3) staining. The M1 regions indicate the cells with the loss of MMP. (C) The percentage of Annexin V-FITC positive cells. (D) The percentage of cells in M1 region. Data are presented as mean ± S.D. in triplicate. (***$p < 0.001$ vs. untreated control; ###$p < 0.001$ between indicated groups).

However, the triple combination of CGA, TC-HT, and LIPEF with the same parameters as above dramatically augmented the population of cells with depolarized mitochondria in comparison to the cells that were untreated or treated with double combination treatment (CGA+-LIPEF, CGA+TCHT, or LIPEF+TC-HT). These observations are consistent with the results of Annexin V-FITC and PI assay (**Fig 5A**), indicating that the mitochondrial dysfunction is implicated in the apoptosis of PANC-1 cells induced by the triple treatment.

## Effect of triple treatment on p53-related proteins

To further investigate the mechanism underlying triple treatment-induced cell cycle arrest and apoptosis, activation of p53-mediated signalling pathway in PANC-1 cells was evaluated using western blot analysis. As shown in **Fig 6A** and **6B**, when compared with the untreated control, each single or double treatment, the triple treatment of PANC-1 cells with CGA (200 μM), TC-HT (10-cycles), and LIPEF (60 V/cm) significantly up-regulated the expression of Bax and down-regulated the expression of Bcl-2, thus resulting in the ratio of Bax/Bcl-2 in favor of apoptosis. Meanwhile, the protein levels of cleaved caspase-9 and PARP cleavage were increased (**Fig 6C** and **6D**). Furthermore, we also found that p53 and p21 protein expression levels in the PANC-1 cells subjected to the triple treatment were significantly higher than those of the control cells or than those of the double combination treatment (CGA+LIPEF, CGA+TCHT, or LIPEF+TC-HT) (**Fig 6E** and **6F**). These data elucidate a pathway of growth arrest and mitochondrial apoptosis in PANC-1 cells that involves up-regulation of p53, p21, Bax, cleaved caspase-9 and PARP as well as down-regulation of Bcl-2.

## The role of ROS production in the triple treatment-induced anticancer effect

We next measured intracellular ROS generation within the cells using a fluorescent probe DHE to reveal that whether oxidative stress is involved in the induction of apoptosis in PANC-1 cells. As shown in **Fig 7A** and **7B**, the mean fluorescence intensity (MFI) of DHE was barely increased after single treatment with CGA (200 μM), TC-HT (10-cycles), or LIPEF (60 V/cm) alone when compared to the control group. However, the CGA-treated cells displayed an elevated ROS production following the exposure to TC-HT or LIPEF. And notably, significantly higher level of ROS was observed in the presence of triple treatment including CGA (200 μM) and dual combination of TC-HT (10-cycles) and LIPEF (60 V/cm) compared to the untreated control or either of the double combination treatments (CGA+LIPEF, CGA+TCHT, and LIPEF+TC-HT). To evaluate the role of ROS generation in cytotoxicity, PANC-1 cells were pretreated with ROS scavenger, *N*-acetyl-cysteine (NAC), for 1 h prior to treatments for 24 h. As shown in **Fig 7C**, the triple treatment-induced cell death was markedly attenuated in the NAC-pretreated group. Therefore, an addition of NAC was able to significantly inhibit cell death signals in the presence of triple treatment, as evidenced by western blot analysis. Furthermore, the induced expression of proteins Bax, p21, and p53 as well as the reduced expression of Bcl-2 protein caused by triple treatment were blocked following the pretreatment with NAC, as shown in **Fig 7D** and **7E**. These results indicate that ROS plays an important role in the action of triple treatment in PANC-1 cells.

## Discussion

Studies have shown that combining two or more agents for combating cancer has the advantages of targeting multiple pathways and lowering administered dosage that reduces cytotoxicity to normal cells [4, 22]. Combination therapies have also been employed with significant effect in treating other critical diseases, such as the triple-drug cocktail for HIV infection [23]

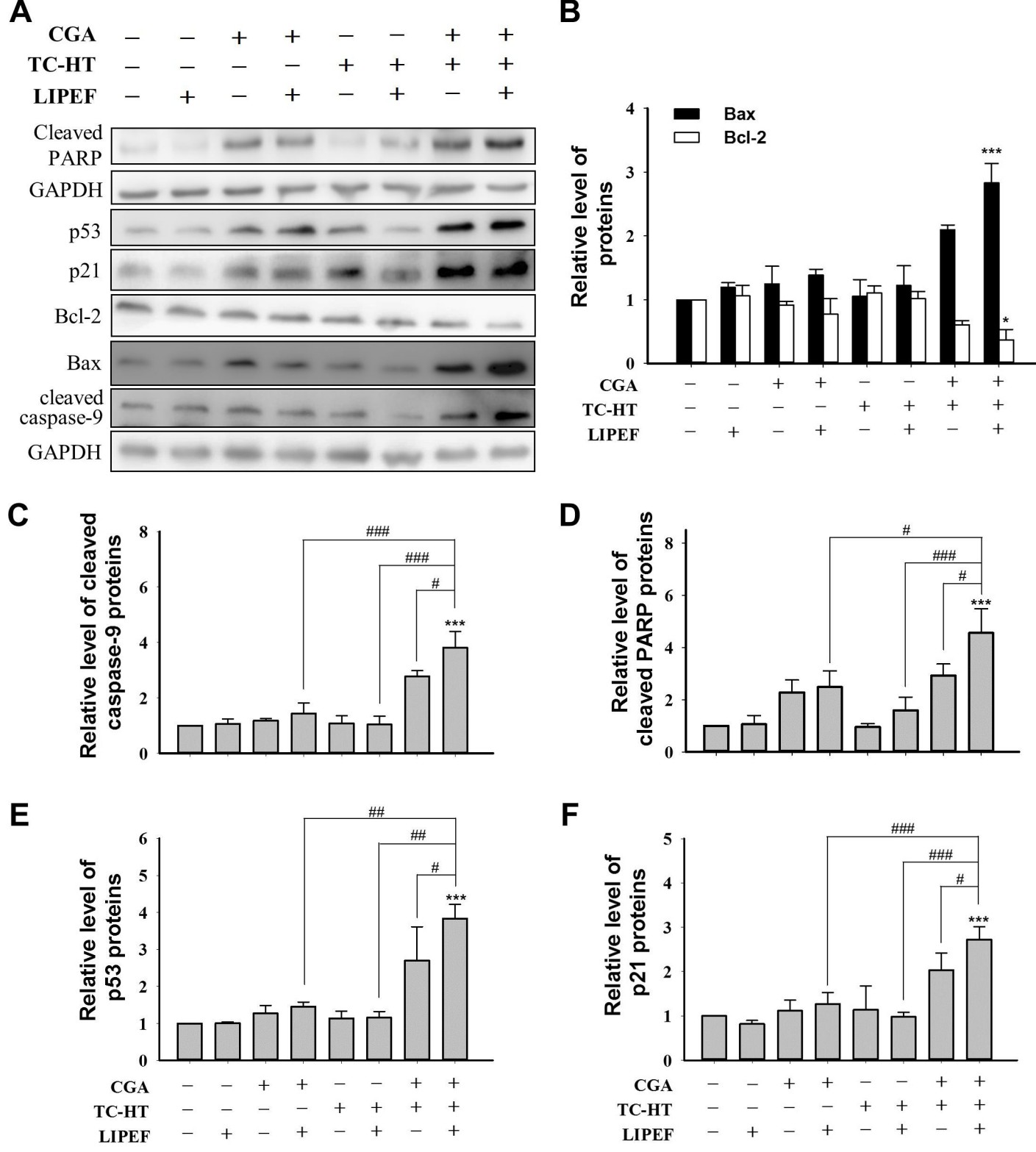

**Fig 6. Effects of single, double or triple combinations of CGA, TC-HT, and LIPEF on the expression of p53-associated proteins in PANC-1 cells.** (A) Representative images of western blotting and the quantification of (B) Bax and Bcl-2, (C) cleaved caspase-9, (D) cleaved PARP, (E) p53, and (F) p21 proteins of the PANC-1 cells treated with CGA (200 μM), TC-HT (10-cycles), and LIPEF (60 V/cm) alone, in double or triple combination treatment for 24 h were examined by western blot analysis. GAPDH was used as internal control. Data are presented as mean ± S.D. in triplicate. (*$p < 0.05$ and ***$p < 0.001$ vs. untreated control; #$p < 0.05$, ##$p < 0.01$, and ###$p < 0.001$ between indicated groups).

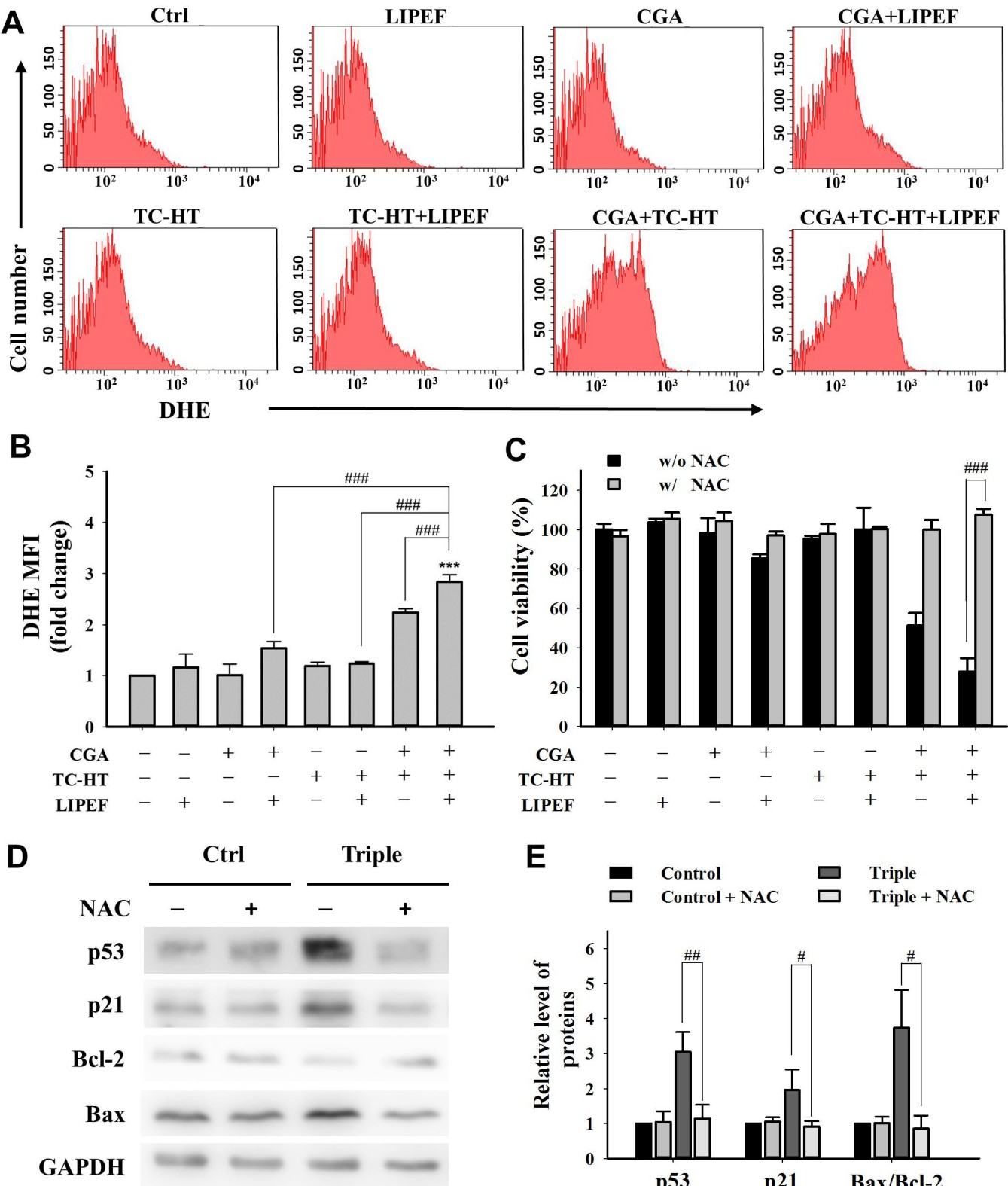

**Fig 7. Combined effects of CGA, TC-HT, and LIPEF on ROS generation in PANC-1 cells.** (A) The DHE (O$_2$$^{•-}$) levels were measured by flow cytometry after treatment with CGA (200 μM), TC-HT (10-cycles), and LIPEF (60 V/cm) alone, in double or triple combination on PANC-1 cells for 24 h. (B) Graph shows the fold change in MFI of DHE relative to control. (C) The PANC-1 cells were treated with the same parameters of CGA, TC-HT, and LIPEF alone or in

combination with or without NAC (5 mM). The cell viability was measured by MTT assay and shown as a percentage of the control. (D) The expression levels of p53, p21, Bax, and Bcl-2 were examined by western blot analysis after the triple treatment with or without NAC (5 mM). GAPDH was used as internal control. The expression levels of these proteins with or without the 5 mM NAC pretreatment were measured and shown in (E), where the expression data were normalized to GAPDH for comparison. Data are presented as mean ± S.D. in triplicate. (***$p < 0.001$ vs. untreated control; #$p < 0.05$, ##$p < 0.01$, and ###$p < 0.001$ between indicated groups).

and the triple therapy for *Helicobacter pylori* [24]. However, the discovery of effective combination treatments needs to explore all possible combination sets of drugs, which is time-consuming and painstaking process [25]. Besides, the combination of different drugs may entail drug competition, discounting the anticancer effect [26]. As a result, the use of physical stimuli has emerged as a remedy to the problem of combination cancer therapy, such as the use of LIPEF, in conjunction with ultrasound and EGCG, put forth in a previous study by our team [8]. In this study, we further propose another modality, combining CGA, TC-HT, and LIPEF, with proof for its significant effect in combating pancreatic cancer.

The study also looks into the anti-proliferative and pro-apoptotic activities of triple treatment against human pancreatic cancer cells. Our results indicate that exposure of the CGA-treated cells to either TC-HT or LIPEF can alter the proliferation of PANC-1 cells. Interestingly, the triple treatment can significantly inhibit cell growth, cutting cell viability to about 20% of the control, comparable to chemotherapy drugs in *in vitro* efficacy [27]. More importantly, the triple treatment selectively kills cancer cells without harming normal cells. The outcome suggests that the synergy effect of the triple combination may result from thermal and electrical enhancement of CGA cytotoxicity. In addition, the study finds that triple treatment significantly induced G2/M phase arrest in the cell cycle progression (**Fig 4**). The tumor suppressor gene p53 has been implicated in various cellular processes such as regulation of cell cycle arrest, apoptosis, and DNA repair [28]. Meanwhile, mutations of p53 are present in approximately half of all human cancers, including pancreatic adenocarcinoma, which results in loss of wild-type p53 function or gain of function that promotes cancer phenotypes [29]. Thus, targeting mutant p53 represents another promising approach to the development of cancer treatment. In the current study, the pancreatic cancer cell line PANC-1, which harbors mutant p53, was treated with triple combination of CGA, TC-HT, and LIPEF to investigate the effect on the expression of p53. As shown in **Fig 6,** triple treatment significantly increased the expression of mutant p53 proteins in PANC-1 cells. The p53-downstream target gene p21, an important cell cycle checkpoint of both G1 and G2/M phases [30], was also examined. P21 can bind to Cdc2-cyclin B complex, which plays a key role in the G2/M progression, and inhibit its kinase activity, thereby blocking cells in G2/M phase [31]. Notably, the level of p21 proteins was also increased after treatment of PANC-1 cells with triple combination (**Fig 6**). Furthermore, a recent study of our team showed that the combination treatment of PANC-1 cells with CGA and TC-HT can induce cell cycle arrest via suppression of Cdc2 and cyclin B1 [7]. Therefore, it is judged the suppression of cell proliferation by triple treatment involves a cell cycle arrest during G2/M phase.

Apoptosis is an autonomous process of cell death, essential for the development and homeostasis of multicellular organisms. The induction of apoptosis as anticancer mechanism has been considered an attractive therapeutic approach, thanks to the minimal cytotoxicity on normal tissues [32]. In the study, our results show that the triple combination of CGA, TC-HT, and LIPEF with certain parameters has a cytotoxic effect on pancreatic cancer PANC-1 cells via triggering apoptotic cell death. Apoptotic pathways can be divided into the extrinsic death-receptor pathway and the intrinsic mitochondrial pathway. **Fig 5** exhibits that triple treatment-induced apoptosis was mediated by a reduction in MMP in PANC-1 cells. The MMP is mainly regulated by interaction between members of Bcl-2 family, and the ratio of

Bax to Bcl-2 is considered as an important upstream checkpoint for mitochondrial pathway. Once the balance of Bax/Bcl-2 ratio was disturbed, the MMP was disrupted, triggering the release of cytochrome *c* into the cytosol, the activation of caspase cascades and ultimately leading to apoptosis [33]. As expected, the triple combination resulted in a decreased expression of Bcl-2 and an increase in that of Bax with a concomitant activation of caspase-9 and cleavage of PARP (**Fig 6B–6D**). P53, also an important activator of the mitochondrial apoptotic pathway, was shown to regulate the expressions of Bcl-2 and Bax *in vitro* and *in vivo* [34]. Thus, the up-regulation of p53 found in our experiments may be the cause for the imbalanced Bax/Bcl-2 ratio in the PANC-1 cells treated with triple treatment (**Fig 6B** and **6E**). Notably, the p53-mediated cytotoxic effect may vary between different cell lines, in particular, the viability of p53-null AsPC-1 cells was not affected by triple treatment (**S1 Fig**). The results suggest the p53 pathway was involved in apoptotic induction in the treatment of PANC-1 cells via the therapy combining CGA, TC-HT, and LIPEF. Previous studies have reported that the high-frequency components of PEF could interact with intermembrane organelles, such as mitochondria and nucleus, when PEF passes through cytomembrane [35, 36]. It indicates that the application of LIPEF could intensify the disturbance of mitochondria triggered by double treatment with CGA and TC-HT, which in turn stimulated the depolarization of MMP. On the other hand, we also applied LIPEF with different frequency to study the synergistic effect on triple combination. Results shown in **S2 Fig** revealed a frequency-dependent reduction in cell viability of PANC-1 following the triple treatment, especially at 2 Hz. Therefore, the repetition frequency and the field intensity of LIPEF may both play a key role in the synergy of actions. Given the fact that the electric field can be modified to focus on targeted area, non-contact LIPEF stimulus featuring simultaneous multi-frequency components to induce beneficial bioeffects appears to be a promising therapeutic approach.

Several studies have indicated that ROS production plays a crucial role in cell cycle progression and apoptosis [37, 38]. On the other hand, it has been found that CGA displays anticancer activity through ROS generation and subsequent induction of apoptosis in cancer cells [39, 40]. In this study, triple treatment significantly stimulated the production of ROS in PANC-1 cells (**Fig 7A** and **7B**), indicating that exposure to TC-HT and LIPEF can boost the cellular ROS level in the CGA-treated cells. Previous studies have indicated that there exists a bidirectional relationship between ROS and p53 [41–43]. ROS can function as a downstream mediator for p53 in triggering mitochondrial apoptosis, as well as an upstream activator for p53 expression. To examine the role of ROS in the anticancer activity of triple treatment, we pretreated the cells with NAC, a ROS scavenger. In response to NAC pretreatment, triple treatment-induced cell death was attenuated (**Fig 7C**). Moreover, the triple treatment-stimulated expression of p53 was reduced in the NAC-pretreated cells (**Fig 7D** and **7E**). The other p53-associated proteins, including Bax and p21, were also decreased by the addition of NAC (**Fig 7D** and **7E**). By contrast, the anti-apoptotic protein Bcl-2 reduced by triple treatment was recovered following NAC treatment. Therefore, ROS produced by triple treatment may act as an inducer for p53 expression. Above all, these findings indicate that CGA in conjunction with both TC-HT and LIPEF can induce the cell cycle arrest and apoptosis through ROS-mediated p53 signalling pathway, as shown in **Fig 8**.

The study first reveals that non-invasive treatment via TC-HT and LIPEF could serve as anticancer agent in synergy with natural compound to selectively cause cancer cell death, with an effect comparable to chemotherapeutic drugs. In *in vivo* experiment, high-intensity focused ultrasound (HIFU) might be a suitable thermal source for TC-HT. Thanks to the ability to fine tune thermal parameters by modulating the heating power and the size of the heated region via multi-focal HIFU, it has been proven to be applicable for mild HT therapy in tumor [44]. This thermal technique is able to control the heated tumor area to only about 4–8 mm [45, 46],

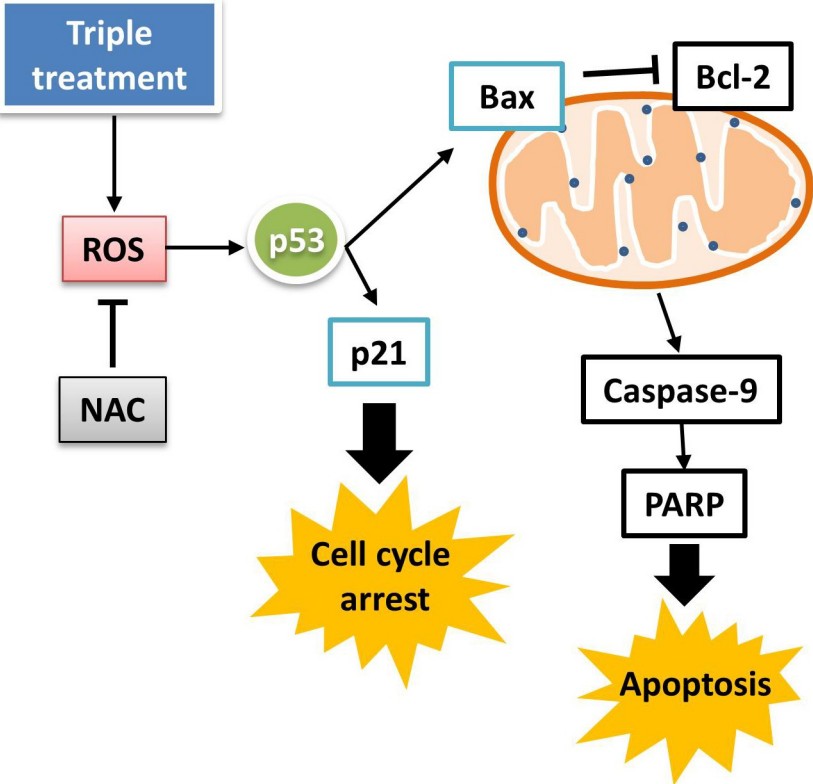

**Fig 8. Schematic illustration for molecular mechanism of cytotoxic activity of the triple treatment on PANC-1 cells with CGA, TC-HT, and LIPEF.**

much smaller than surrounding bulk tumor/tissue mass, as a result of which heat can quickly dissipate in the passive cooling environment of human body. In operation, the size of scan spot (the heated area) and the heating intensity adopted by multi-focal HIFU in the scanning mode can be tunable so as to achieve effective thermal dissipation necessary for application of TC-HT *in vivo*. As for other cycling parameters that require a higher thermal dissipation rate, active cooling devices can be incorporated for pre-cooling [47]. Therefore, the combination modality of TC-HT and LIPEF along with natural compound CGA appears to be a potent approach for treatment of pancreatic cancer with the potential of improving the life quality of such patients. This new method provides a means of mild anticancer treatment by alleviating the side effects associated with prolonged thermal exposure and direct contact with electrodes. However, the synergetic effect of non-invasive thermal and electrical stimuli on CGA may vary among different cancer cells. Therefore, further studies are needed to establish its benefits in other cancer cell lines.

In summary, the study puts forth a unique triple treatment combining CGA, TC-HT, and LIPEF, which significantly inhibited the growth of PANC-1 cells via ROS-mediated G2/M arrest and mitochondrial apoptosis. Our results indicate that the molecular mechanisms underlying the triple treatment-induced cell cycle arrest and apoptosis in PANC-1 cells are found to be associated with the up-regulation of mutant p53. Activation of p53 signalling pathway can lead to the increased expression of p21 and Bax as well as the suppression of Bcl-2, accompanied by cell cycle arrest and mitochondrial dysfunction along with the activation of caspase-9 and PARP, eventually triggering apoptosis. Overall, this paper points to the potential of triple treatment as an alternative approach for anticancer treatment.

## Supporting information

**S1 Fig. The cell viability of AsPC-1 cells exposed to CGA (200 μM), TC-HT (10-cycles), and LIEF (60 V/cm) alone or in combination for 24 h.** Data are presented as mean ± S.D. in triplicate.
(TIFF)

**S2 Fig. The cell viability of PANC-1 cells exposed to CGA (200 μM), TC-HT (10-cycles), and 60 V/cm LIEF (2, 20, and 100 Hz) alone or in combination for 24 h.** Data are presented as mean ± S.D. in triplicate.
(TIFF)

**S1 File. Original blot images of western blot analysis.**
(ZIP)

**S2 File. Raw data of cell viability and colony formation results.**
(PDF)

**S3 File. Raw data of flow cytometry result.**
(PDF)

## Acknowledgments

We would like to thank Technology Commons in College of Life Science, National Taiwan University for use of flow cytometry system, and the staff of the imaging core at the First Core Labs, National Taiwan University Hospital for technical assistance.

## Author Contributions

**Conceptualization:** Chih-Yu Chao.

**Data curation:** Chueh-Hsuan Lu, Yu-Yi Kuo.

**Formal analysis:** Chueh-Hsuan Lu, Yu-Yi Kuo, Guan-Bo Lin, Wei-Ting Chen, Chih-Yu Chao.

**Funding acquisition:** Chih-Yu Chao.

**Investigation:** Chueh-Hsuan Lu, Yu-Yi Kuo, Guan-Bo Lin, Chih-Yu Chao.

**Project administration:** Chih-Yu Chao.

**Supervision:** Chih-Yu Chao.

**Validation:** Chueh-Hsuan Lu, Yu-Yi Kuo.

**Writing – original draft:** Chueh-Hsuan Lu, Chih-Yu Chao.

**Writing – review & editing:** Chih-Yu Chao.

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
