## [Decision Letter · Decision Letter 0]

1 Oct 2019

PONE-D-19-23209

Application of non-invasive low-intensity pulsed electric field with thermal cycling-hyperthermia for synergistically enhanced anticancer effect of chlorogenic acid on PANC-1 cells

PLOS ONE

Dear Dr. Chao,

Thank you for submitting your manuscript to PLOS ONE. After careful consideration, we feel that it has merit but does not fully meet PLOS ONE’s publication criteria as it currently stands. Therefore, we invite you to submit a revised version of the manuscript that addresses the points raised during the review process.

Please, address the reviewers' points.

We would appreciate receiving your revised manuscript by Nov 15 2019 11:59PM. To enhance the reproducibility of your results, we recommend that if applicable you deposit your laboratory protocols in protocols.io, where a protocol can be assigned its own identifier (DOI) such that it can be cited independently in the future. For instructions see: http://journals.plos.org/plosone/s/submission-guidelines#loc-laboratory-protocols

We look forward to receiving your revised manuscript.

Kind regards,

Ferenc Gallyas, Jr., Ph.D., D.Sc.

Academic Editor

PLOS ONE

Journal Requirements:

Additional Editor Comments (if provided):

Reviewers' comments:

Reviewer's Responses to Questions

**Comments to the Author**

1. Is the manuscript technically sound, and do the data support the conclusions?

Reviewer #1: Partly

Reviewer #2: Yes

2. Has the statistical analysis been performed appropriately and rigorously? 

Reviewer #1: Yes

Reviewer #2: Yes

3. Have the authors made all data underlying the findings in their manuscript fully available?

Reviewer #1: Yes

Reviewer #2: Yes

4. Is the manuscript presented in an intelligible fashion and written in standard English?

Reviewer #1: Yes

Reviewer #2: No

5. Review Comments to the Author

Reviewer #1: The paper proposes a novel treatment modality includes non-invasive treatment CGC, coupled with thermal-cycling hyperthermia (TC-HT) and low-intensity pulsedelectric field (LIPEF). The triple combination leads to excessive apoptotic cell death with known signatures for apoptosis in Panc-1 pancreatic cancer cells. Although MS includes non-malign cell data, due to genetic heterogenety of pancreatic tumors, other cell lines carry different mutations for p53 or KRas is missing. MS skips the mutational loss of functional p53 status of Panc-1 cells in discussion part. Therefore discussion part should be revised according to p53 burden in tumors and effect on therapy. There are number of typo errors in the text.

Reviewer #2: Title: "non-invasive low-intensity pulsed electric field" - The description of PEF is lacking in the M&M and why use it in their studies. It is noteworthy that there are electric fields being larger fields (1.8 and 1.26 kV/cm) can produced a high drug intake rate during the first 10 ms that can decrease significantly thereafter. On the other hand, the intake rate during the first 10 ms for the two smaller fields (0.63 and 0.45 kV/cm) maybe undetectable and may gradually increased afterward. The rates for these two fields can be similar to one another until the end of the analysis. In addition, the optimal PEF levels for in vitro tests need to be described in detail; e.g., how many pulses at 1 kHz and 0.45 kV/cm to be nonlethal to normal cells but effectively kill cancer cells.

Abstract: "In this study, we propose a novel non-invasive treatment calling for exposure of cells to CGA, coupled with thermal-cycling hyperthermia (TC-HT) and low-intensity pulsed electric field (LIPEF)." Why use non-invasive here? No definition for CGA here. The authors published a previous article (PLoS One. 2019; 14(5): e0217676.Published online 2019 May 31. doi: 10.1371/journal.pone.0217676) using CGA and TC-HT treatment of PANC1 cells. There is no rationale for using LIPEF in this study.

Introduction: As written, it does not provide a rationale for the study. The authors have published a paper on CGA and TC-HT which needs to be articulated at the beginning with the findings. Then, why use LIPEF as a combination treatment with CGA and TC-HT in this study. This section is poorly written. For example, "We have recently shown that the thermal cycling-hyperthermia (TC-HT) has an anticancer synergy via coupling with the polyphenols, chlorogenic acid (CGA) and epigallocatechin gallate (EGCG) in combating human pancreatic cancer PANC-1 cells [7], thanks to the effect of the cell cycle arrest and the induction of mitochondria-mediated apoptosis, which were not found in the separate treatments."

Discussion: Poorly written. "Unlike alternating current signal stimulus, non-invasive LIPEF can be viewed as the composition of many sinusoidal subcomponents with multiple frequencies [30]. When the electric pluses enter and interact with a biological sample, we propose that the LIPEF is dispersed electrically to multiple frequencies due to the encounter with biological dielectrics. Thus, the LIPEF’s multiple frequencies, caused by electrical dispersion in biological dielectrics, can simultaneously interact with molecules, proteins, organelles, and cells to induce bioeffects during the LIPEF exposure. The study points out that the application of LIPEF could intensify the disturbance of mitochondria triggered by double treatment with CGA and TC-HT, which in turn stimulated the depolarization of MMP." Unfortunately, the description of LIPEF is lacking in the study which needs to be addressed in detail.

6. PLOS authors have the option to publish the peer review history of their article (what does this mean?). If published, this will include your full peer review and any attached files.

Reviewer #1: No

Reviewer #2: No

---

## [Author Response · Author response to Decision Letter 0]

14 Nov 2019

We are grateful to the reviewers for their comments and suggestions in this manuscript. In the revised manuscript, we have adopted the reviewers’ comments and suggestions, and believe the manuscript now is in a final form suitable for publication.

---

## [Editor Report · Decision Letter 1]

20 Nov 2019

PONE-D-19-23209R1

Application of non-invasive low-intensity pulsed electric field with thermal cycling-hyperthermia for synergistically enhanced anticancer effect of chlorogenic acid on PANC-1 cells

PLOS ONE

Dear Dr. Chao,

Thank you for submitting your manuscript to PLOS ONE. After careful consideration, we feel that it has merit but does not fully meet PLOS ONE’s publication criteria as it currently stands. Therefore, we invite you to submit a revised version of the manuscript that addresses the points raised during the review process.

PLOS ONE now requires that submissions reporting blots or gels include original, uncropped blot/gel image data as a supplement or in a public repository. Although not uncropped, the images for PARP and corresponding GAPDH would have satisfied PLOS ONE's request. However, the rest of the blot images often as closely cropped as in the figures only presented on a larger clipboard. Please provide images for original blots of  Bcl-2, Bax, their corresponding GAPDH and all +NAC blots that meet PLOS ONE's requirement.

We would appreciate receiving your revised manuscript by Jan 04 2020 11:59PM. To enhance the reproducibility of your results, we recommend that if applicable you deposit your laboratory protocols in protocols.io, where a protocol can be assigned its own identifier (DOI) such that it can be cited independently in the future. For instructions see: http://journals.plos.org/plosone/s/submission-guidelines#loc-laboratory-protocols

We look forward to receiving your revised manuscript.

Kind regards,

Ferenc Gallyas, Jr., Ph.D., D.Sc.

Academic Editor

PLOS ONE

---

## [Author Response · Author response to Decision Letter 1]

6 Dec 2019

We would like to thank editor for the reminder. We have provided images for original blots that meet PLOS ONE's requirement, and believe the manuscript should now be acceptable for publication.

---

## [Editor Report · Decision Letter 2]

10 Jan 2020

Application of non-invasive low-intensity pulsed electric field with thermal cycling-hyperthermia for synergistically enhanced anticancer effect of chlorogenic acid on PANC-1 cells

PONE-D-19-23209R2

Dear Dr. Chao,

We are pleased to inform you that your manuscript has been judged scientifically suitable for publication and will be formally accepted for publication once it complies with all outstanding technical requirements.

With kind regards,

Ferenc Gallyas, Jr., Ph.D., D.Sc.

Section Editor

PLOS ONE
---

## [Editor Report · Acceptance letter]

16 Jan 2020

PONE-D-19-23209R2 

Application of non-invasive low-intensity pulsed electric field with thermal cycling-hyperthermia for synergistically enhanced anticancer effect of chlorogenic acid on PANC-1 cells 

Dear Dr. Chao:

I am pleased to inform you that your manuscript has been deemed suitable for publication in PLOS ONE. Congratulations! Your manuscript is now with our production department. 

With kind regards,

on behalf of

Dr. Ferenc Gallyas, Jr. 

Section Editor

PLOS ONE